# CICD-Coder: Chinese EMRs Based ICD Coding With Multi-axial Supported Clinical Evidence

## Abstract

Although automatic ICD coding has achieved some success in English, there still exist significant challenges for the Chinese electronic medical records(EMRs) based ICD coding task. The first problem is the difficulty of extracting disease code-related information from Chinese EMRs due to the concise writing style and specific internal structure content of EMRs. The second problem is that previous methods have not exploited the disease-based multi-axial knowledge and are neither associated with the corresponding clinical evidence, resulting in inaccuracy in disease coding and lack of interpretability.

In this paper, we develop a novel automatic ICD coding framework CICD-Coder for the Chinese EMRs-based ICD coding task. In the presented framework, we first investigate the multi-axes knowledge (crucial for the ICD coding) of the given disease and then retrieve corresponding clinical evidence for the disease-based multi-axes knowledge from the whole content of EMRs. Finally, we present an evaluation module based on the masked language modeling strategy to ensure each knowledge under the axis of the recommended ICD code is supported by reliable evidence. The experiments are conducted on a large-scale Chinese EMRs dataset collected from varying hospitals and the results verify the effectiveness, reliability, and interpretability of our proposed ICD coding method.

## 1 Introduction

The International Classification of Diseases (ICD) was developed by the World Health Organization (WHO), which converts disease diagnosis descriptions and other health problems into an alphanumeric coding system. ICD codes are widely accepted and used in many countries for clinical research, health care management, and insurance compensation. The design principle of the ICD coding system is very different from the diseases themselves, therefore only the people who have passed the ICD coding exam can assign the appropriate ICD codes based on the patient's medical records independently. In China, these professional ICD coders are few, which inspired several researchers from technology companies and universities to explore automatic ICD coding methods.

A series of competitive frameworks for automatic ICD coding is based on the Bert method (Ji et al. (2021); Zhang et al. (2020); Pascual et al. (2021)), in which each ICD code is associated with a unique entity representation, and the automatic coding task is transformed into a multi-label classification across fine-tuning. Cao et al. (2020) proposed Clinical-Coder, which exploits a dilated convolutional attention network with an $n$-gram matching mechanism to capture semantic features for non-continuous words and continuous n-gram words for automatic ICD coding. Yuan et al. (2022) proposed the MSMN method, which collected synonyms of every ICD code from UMLS and designed a multiple synonyms matching network to leverage synonyms for better code classification. Yang et al. (2023) presented an autoregressive generation with a prompt framework for ICD coding by designing a mask objective for the clinical language model. Yang et al. (2022) utilized prompt-tuning in the context of ICD coding tasks by adding a sequence of ICD code descriptions as prompts in addition to each clinical note as KEPT LM input.

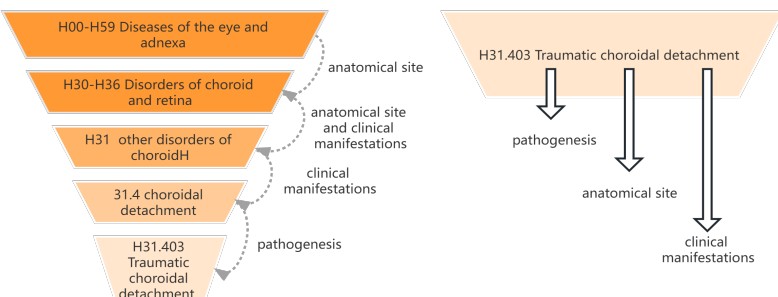

Figure 1: A coarse-to-fine classification process under multi-axes of ICD code H31.403 (which is from the Chinese medical insurance version). Each part of the code is the specific knowledge of the corresponding axis.

Although automatic ICD coding has received much attention, there is little work that has focused on the difficulties and realistic requirements for Chinese electronic medical records. We point out the following two issues that have not been paid attention to and solved.

The first problem is the difficulty of extracting disease code-related information from Chinese EMRs due to the concise writing style and specific internal structure content of EMRs. The automatic ICD coding takes diagnosis-related descriptions, as well as other key information like symptom manifestations and auxiliary examination results as input. In English EMRs, diagnosis-related information is described in a short sentence, such as in the MIMIC-III dataset (Johnson et al. (2016)). However, the diagnosis is usually described by a phrase consisting of several words in Chinese EMRs. Furthermore, the coding-dependent information is generally recorded in the discharge summary of English EMRs, as researchers truncate discharge summaries of the MIMIC-III dataset at 4,000 words as input(Xie et al. (2019), Yuan et al. (2022)). Although there is a discharge summary in the EMR, the content is only a summary of the hospitalization process in several paragraphs, which is insufficient to support automatic coding. To substantiate the aforementioned statement, we have listed statistical information in Table 1, which presents the length statistics of diagnostic descriptions and discharge summaries in a Chinese and English EMRs dataset. The information required for coding is scattered in every possible corner of the EMR. For example, if we want to determine whether a tumor is malignant, we need to look for descriptions in the pathology report. If we want to determine whether to assign an intestinal infectious disease code, we need to check the laboratory test results. In a word, the diagnosis description is too short, and coding-dependent information is scattered everywhere, which makes the previous methods not practical for Chinese EMRs.

Table 1: Statistics of the Chinese and MIMIC-III datasets, including the mean, variance, and median of the number of words per diagnosis, and the number of words per discharge summary respectively. For the Chinese dataset, the words were obtained through the jieba word segmentation tool.[2]

| Chinese dataset | Mean | Variance | Median |
|---|---|---|---|
| # Words per diagnosis | 3.74 | 3.99 | 3 |
| # Words per discharge summary | 731.80 | 104742.07 | 706 |
| MIMIC-III dataset | Mean | Variance | Median |
| # Words per diagnosis | 14.73 | 276.91 | 11 |
| # Words per discharge summary | 1600.36 | 600303.20 | 1466 |

The second problem is that previous methods have not exploited the disease-based multi-axial knowledge and are neither associated with the corresponding clinical evidence, resulting in inaccuracy in disease coding and lack of interpretability. For example, Clinical-Coder (Cao et al. (2020)) claimed to concentrate on explaining the reason why each ICD code is to be predicted. However, this method can only find several words in the medical record similar to the ICD code name as an explanatory basis. In fact, such explicit clues are rare. In practical situations, users may have low fault tolerance and acceptance rates for the kind of automated ICD coding system without reliable evidence. Even a small error can greatly impact users' trust and ultimately lead to the system not being accepted and finally deprecated.

---

[2]Here is the code link for jieba word segmentation tool: `https://github.com/fxsjy/jieba`

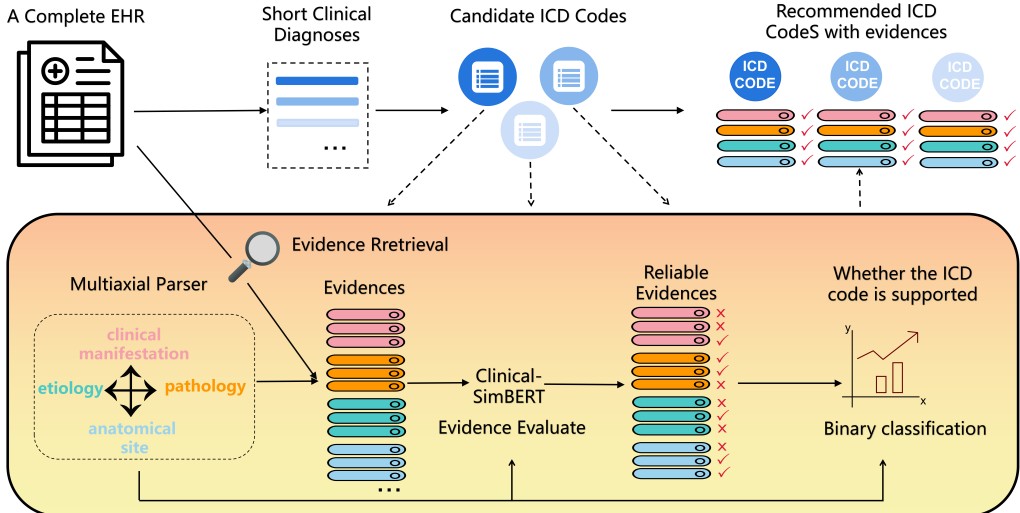

Figure 2: The general framework of the proposed method.

The ICD coding system classifies health status according to the principle of coarse to fine under multiple classification axes. There are four kinds of classification axes: etiology, anatomical site, pathology, and clinical manifestations. We show a coarse-to-fine extraction process of ICD code H31.403 (which is from the Chinese medical insurance version) in Figure 1, and each part of the code is the specific knowledge of the corresponding axis. Inspired by the above observation and evidence-based medicine (Sackett (1997)), we proposed a knowledge-evidence ICD coding schema, which means the multi-axial knowledge of the recommended ICD code should be supported by the textual evidence from the corresponding EMR. Such necessary and reliable evidence guarantees the correctness of the recommended code.

To address the two issues mentioned above, we developed CICD-Coder: Chinese EMRs-based ICD coding with multi-axial supported clinical pieces of evidence, which is shown in Figure 2. This framework is inspired by the coarse-to-fine classification strategy under multi-axes knowledge of the ICD codes system and the theory of evidence-based medicine (Sackett (1997)). Specifically, we first extract the diagnosis list from the Chinese EMR. Then we find the most likely candidate ICD codes for each diagnosis for efficiency. We use a multi-axial parser to divide the knowledge under the axis for each candidate code. We further use an evidence retrieval module based on prior knowledge from ICD coders, to find out the supporting text description in the EMR for knowledge under each axis of the candidate code. To ensure the reliability of the evidence retrieved, we propose a Clinical-Simbert model based on the similarity computation between each piece of evidence and corresponding knowledge under the axis. After obtaining the reliable evidence set, we need to judge whether the ICD code is fully supported by the evidence set and worth recommending. Inspired by the prompt tuning method which bridges the gap between pre-training and fine-tuning downstream tasks (Schick & Schütze (2020); Liu et al. (2021)), and is demonstrated can get better performances. We employ the prompt-tuning method by designing a template that combines evidence under all axes and the candidate ICD code. Then we convert the binary classification task into a masked language modeling problem to conduct supervised training. Finally, we obtain the recommended ICD code for all diagnoses and the corresponding evidence set for each diagnosis.

We summarize the contribution of this paper as follows:

- We point out the differences between Chinese EMRs and English EMRs in extracting ICD coding-dependent information. The diagnosis in Chinese EMRs is just several words, and other ICD coding-depend information is scattered in every corner of the medical record. Therefore, the previous ICD automatic coding method focused on English EMRs is not applicable to Chinese EMRs.

- Inspired by the coarse-to-fine classification strategy under multi-axes knowledge of the ICD codes system and the theory of evidence-based medicine, we proposed an evidence retrieve module to find out the textual description in the EMR supporting the knowledge under the axis of each candidate ICD code, and further evaluate the quality of evidence to ensure its reliable by proposed Clinical-Simbert model;

- We convert the binary classification task of whether the candidate ICD code can be fully supported by the evidence set into a masked language modeling problem by designing a template that combines the evidence set and the candidate ICD code. We conducted ICD code prediction experiments on the actual Chinese dataset collected and obtained state-of-the-art performance. We finally present the assessment results by ICD coders to verify the reliability and interpretability of the evidence.

## 2 METHODS

### 2.1 TASK DEFINITION

Recently, reducing a multi-label to a set of binary classification problems has proved to be competitive to more sophisticated multi-label classification methods, and still achieves state-of-the-art performance for many loss functions (Wever et al. (2020)). Inspired by this, We take the automatic ICD coding task as a set of binary classification tasks based on diagnoses and the overall EMR.

Given an EMR, we first find out the Top $N$ candidate ICD codes for each diagnosis. Then we find out the supporting evidence set for knowledge under the multi-axis of each candidate ICD code by evidence retrieving and evaluating. We determine whether the ICD code is fully supported by the evidence set. Specifically, we take $i$th candidate ICD code description $c_i$ and its corresponding evidence set $E_i = \{e_1, e_2, ..., e_K\}$ organized in template form as input, which $K$ indices the maximum number of evidence. And assign a binary label $y_i \in \{0, 1\}$ for the $i$th candidate ICD code in the label space $Y$, where 1 means that the ICD code is sufficiently supported by the evidence and to be recommended, 0 is the opposite.

### 2.2 RETRIEVE AND EVALUATION OF SUPPORTING EVIDENCE UNDER MULTIPLE AXES

#### 2.2.1 OBTAINING CANDIDATE CODES FOR EACH DIAGNOSIS

We gather all diagnoses from the medical record. We use the edit distance (Marzal & Vidal (1993)), tfidf, and word2vec-based (Mikolov et al. (2013)) similarity computation methods to capture Top $N$ candidate codes among all the ICD codes, which are related to each diagnosis at the character, discrimination, and semantic level respectively.

#### 2.2.2 GREEDY EVIDENCE RETRIEVE BASED ON PRIOR KNOWLEDGE

The prior knowledge we utilized was obtained through discussions with experienced professional ICD coders. Specifically, the ICD coding system categorizes all codes into 212 categories at the three-digit level Buck (2016). Through our discussions, we determined that each category focuses on similar likely appearance locations of the supporting evidence in the medical record. We summarized a total of 88 possible locations where evidence may appear in Chinese EMRs. Furthermore, we asked the ICD coders to annotate the possible locations of each code category, specifically focusing on the evidence that may appear. This process resulted in the formation of a prior information table. For example, if we want to determine whether a tumor is malignant, we need to look for descriptions in the pathology report. To provide a clearer understanding of this prior knowledge, we have included a small excerpt from the full table in the appendix.

After obtaining the set of candidate ICD codes for each diagnosis, we extract knowledge words under multi-axes of each candidate code by a pre-trained parser as keywords for evidence retrieval. In addition, the MSMN method (Yuan et al. (2022)) has verified that rich synonym descriptions can be more effective in ICD coding, therefore we use synonyms for these keywords as a supplement. As well as the evidence that contains more keywords is considered to be more representative and reliable. Based on these, we sort all possible combinations of keywords in descending order according

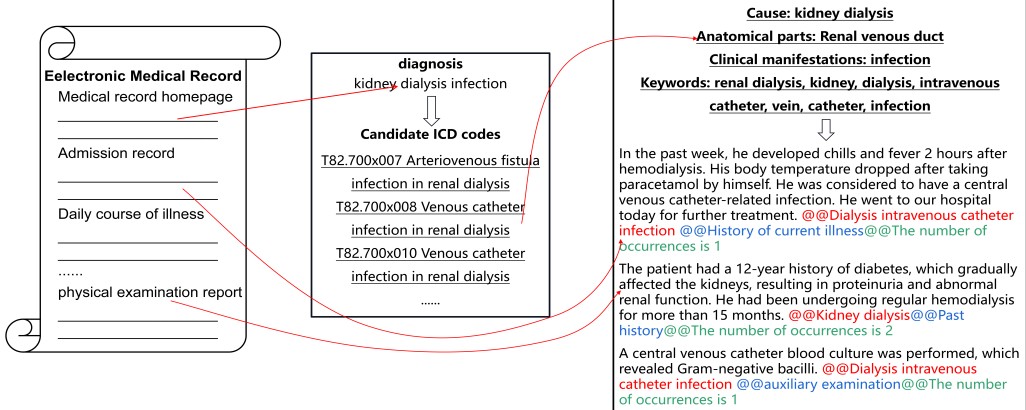

Figure 3: A real example to show the candidate codes of one diagnosis and the reliable evidence under multi-axes for one of the candidate codes (also the correct ICD code), the example is from a real Chinese EMR. Each piece of evidence is followed by a few "@@" symbols for more information helpful for inference, including keywords, the appearing field in the medical record, and times of appearance (The Chinese version of this figure is in the Appendix).

to the number of keywords and retrieve the sentence containing them in all likely appearance locations. We replaced the sentence containing fewer keywords with more evidence based on a greedy strategy. Once the supporting evidence for a keyword has been retrieved, the retrieval will not continue based on this keyword. We stopped retrieving until we traversed all keywords. Supported by prior information, the overall process is fast and efficient.

### 2.2.3 Reliability assessment of evidence

To ensure that the evidence containing the combination of keywords is a proper context for supporting the knowledge of the corresponding axis, We assess the reliability of the evidence by an evidence-scoring model named Clinical-Simbert. Specifically, we first pre-trained a sentence embedding model by the Simbert framework (Su (2020)), which performed unsupervised contrast learning based on a large number of sentence pairs from Chinese EMRs. For obtaining training data, we take sentence pairs that contain more than 60% of the same medical entities as positively related sentence pairs, and randomly sample pairs from EMRs in different departments as negatively related sentence pairs. Based on this model, We further conduct contrastive learning based on a knowledge-evidence dataset. Specifically, we use the correct ICD code, keywords, and retrieved evidence in the training set as positively related pairs, and random sample the evidence of wrong candidate ICD codes as negatively related pairs. The Clinical-Simbert model after two rounds of training can judge the reliability of the evidence for the knowledge under the axes of ICD codes.

After obtaining a relevance score by Clinical-Simbert, we removed the sentence with a similarity score below the threshold value, and the remaining evidence is considered reliable. As shown in Figure 3, we give a real example to show the candidate codes of one diagnosis and the reliable evidence under multi-axes for one of the candidate codes (also the correct encoding result), the example is from a real Chinese EMR. Each piece of evidence is followed by a few "@@" symbols for more useful information, which will be described in detail in the next section.

### 2.3 Inference Based on Prompt-tuning method

Inspired by the prompt tuning method which bridges the gap between pre-training and fine-tuning downstream tasks (Schick Sch utze (2020); Liu et al. (2021)), and is demonstrated can get better performances. We convert the binary classification task of whether the candidate ICD code can be fully supported by the evidence set into a masked language modeling problem. Whether the method works depends largely on the design of the template. So we will introduce the template design and prediction in this section. The inference framework is shown in Figure 4.

### 2.3.1 TEMPLATE DESIGN

Through our observations of Chinese electronic medical records and communication with many professional ICD coders and clinicians, we have summarized two principles for writing Chinese EMRs, which will help me design the template for prompt tuning.

**Principle1: The source of the evidence limits what the evidence can describe and how credible it is.**

For example, the past history records the patient's condition before the current hospitalization, whereas the present history records the patient's condition about the current hospitalization. Therefore, if the evidence about the patient's current health status comes from the past history, its reference value is much lower than that from the present history. In addition, the content of the present history comes from the doctor's summary of the patient's own description of his or her health status, so the content is subjective and not very reliable. This is because each patient has a different ability to perceive and express the disease, and the professional level of doctors also varies. Take patients as an example, some people suffer from heart disease and the corresponding symptom is pain in the back of the heart. However, the patient did not expect that he would have a heart-related disease, so he guessed and then described back pain. In contrast, the evidence from auxiliary examinations is considered to be more objective and accurate. Based on this, the source of the evidence implies very important information that needs to be added to the template.

**Principle2: The number of repeating times of one piece of evidence in the Chinese EMR determines its importance** The patient's essential diagnostic basis and auxiliary examination results support the formulation of each treatment plan and the writing logic of medical records, which are often repeated several times in the medical record. Based on this, we use the $k$-means method to remove duplicate or highly similar evidence from the evidence set retrieved under a set of keywords. We count the repeating times of one evidence and add it to the template, to reflect the importance of the evidence. Finally, because each piece of evidence is to prove the knowledge of the cor-

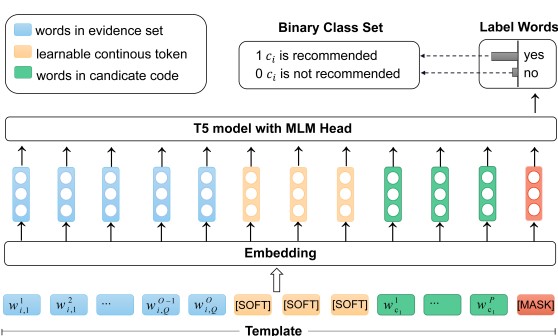

Figure 4: The inference framework based on prompt-tuning.

responding axis, we emphasize them by adding the keywords behind the evidence. All additional information is appended to the evidence and separated by @@ symbols, as shown in Figure 3.

With the above principles, we give the formal definition of the template. Given the $i$th candidate ICD code description $c_i = \{w_{ci}^1, w_{ci}^2, ..., w_{ci}^P\}$ composed of words, $w_{ci}^p$ indices $p$th word of candidate ICD code $c_i$ and $P$ indices the maximum number of words. The corresponding evidence set is $E_i = \{e_i^1, e_i^2, ..., e_i^Q\}$, which $Q$ indices the maximum number of evidence. Each piece of evidence $e_i^q$ is composed of four parts, that is, 1) the sentence is retrieved from the clinical record; 2) the combination of keywords used for retrieval; 3) the field of the sentence originates from; 4) the repeating times if the evidence in the whole Chinese EMR. Because these parts are all composed of words, we represent them as a whole by $\{w_{i,q}^1, w_{i,q}^2, ..., w_{i,q}^O\}$ in which $w_{i,q}^o$ indicate the $o$th word in $q$th evidence of $i$th candidate ICD code, and $O$ indices the maximum number of words. We designed the template as follows:

$$T = [CLS] \; E_i \; [soft] \; [soft] \; [soft] \; c_i \; [soft] \; [mask]$$

In which [soft] indices a learnable continuous embedding, which is a trick to eliminate the trouble of designing prompts and can get better results than designing templates manually proposed by (Li & Liang (2021); Lester et al. (2021)). We use a [mask] token for label prediction, indicating a binary classification label prediction.

### 2.3.2 EVIDENCE-BASED ICD CODES PREDICTION

Raffel et al. (2020) proposed T5 model, which is a unified framework that converts all text-based language problems into a text-to-text format. We employed this model as a text encoder, which embeds input tokens to embedding space:

$$T = [e_{w_{i,1}^1}, ..., e_{w_{i,Q}^O}, e_{soft}, e_{soft},$$
$$e_{soft}, e_{w_{ci}^1}, ..., e_{w_{ci}^P}, e_{soft}, e_{mask}] \tag{1}$$

where $e_{w_{i,1}^1}$ is word embedding of $w_{i,1}^1$, $e_{soft}$ is the embedding of [soft], which is initialized following the T5 model. $e_{mask}$ is the embedding of [mask], which is initialized by the [mask] token by T5 model. We've deleted [cls] for brevity. T5 model then encodes $T$ to achieve the hidden states:

$$T = [h_{w_{i,1}^1}, ..., h_{w_{i,Q}^O}, h_{soft}, h_{soft},$$
$$h_{soft}, h_{w_{ci}^1}, ..., h_{w_{ci}^P}, h_{soft}, h_{mask}] \tag{2}$$

To convert the binary classification task into a masked language modeling problem, we calculate the probability of filling the token to the [mask] position in the template. Let $\mathcal{V}$ and $\mathcal{Y}$ indices a set of label words and class set respectively. An injective mapping function $\phi : \mathcal{Y} \rightarrow \mathcal{V}$ is needed to associate the label set with the set of label words, which is also called "verbalizer" in some recent papers (Hu et al. (2021); Li & Liang (2021)). We define a sample verbalizer by setting $\mathcal{V}_1$ = "yes", $\mathcal{V}_0$ = "no". With the verbalizer, the probability distribution of $\mathcal{Y}$ can be obtained by the probability distribution of $\mathcal{V}$ at the [mask] position, which is formalized as follows:

$$p(y|T) = p([mask] = \phi(y)|T) \tag{3}$$

## 3 EXPERIMENTS

### 3.1 DATASETS

We verify the performance of the proposed method on the Chinese dataset. We collected patients' medical records from different hospitals in four cities. We use the Chinese medical insurance version of the ICD coding system, which contains 33243 ICD codes. We list the statistics of the dataset in Table 2, including the number of Chinese EMRs, the average length of the overall EMR, the number of all ICD codes, the number of all candidate ICD codes, and the recall rate of the candidate ICD codes to the correct ICD codes for the training set, development set, and test set respectively. We also show the mean, variance, and median values of the candidate codes of each diagnosis, the evidence of each candidate code, and tokens of each evidence respectively in Table 3.

### 3.2 BASELINES AND METRICS

We choose the following competitive methods as baselines. We measure the results using macro F1, accuracy, and precision@5 for the Chinese dataset.

**LAAT&jointLAAT**(Vu et al. (2020)) proposed a label attention model for ICD coding, which can handle both the various lengths and the interdependence of the ICD code-related text fragments.

**T5** Raffel et al. (2020) proposed the T5 model, which is a unified framework that converts all text-based language problems into a text-to-text format.

**MSMN** Yuan et al. (2022) proposed a multiple synonyms matching network to leverage synonyms for better code representation learning, and finally help the code classification. MSMN is the state-of-the-art method for automatic ICD coding.

### 3.3 IMPLEMENTATION DETAILS

During the experiment, we set the number of candidate codes for each diagnosis to 50. Under this setting, the recall rate of both the training set, the development set, and the test set exceeds 95%. If the number is increased to 100, the recall rate can exceed 98%, we chose 50 for efficiency. We used edit distance, tfidf, and similarity calculation methods to find top1-top30, top31-top40, and top41-top50 candidate codes for each diagnosis. The similarity calculation needs the embedding of

Table 2: Statistics of the Chinese dataset collected, including the total number of Chinese EMRs, the average length of the overall EMR, the number of all ICD codes, the number of all candidate ICD codes and the recall rate of the candidate ICD codes to the correct ICD codes for the training set, development set, and test set respectively.

| Chinese dataset | Train | Dev | Test |
|---|---|---|---|
| Total # EMR. | 70,318 | 8,689 | 8,790 |
| Avg # Chinese words per EMR. | 83317.35 | 81246.56 | 82328.16 |
| Total # ICD codes. | 2,238 | 2,198 | 2,207 |
| Total # candidate ICD codes. | 11,080 | 10,954 | 11,023 |
| Recall rate | 96.58% | 95.43% | 95.55% |

Table 3: Statistics of the Chinese dataset, including the mean, variance, and median value of the candidate codes of each diagnosis, the evidence of each candidate code, and tokens of each evidence.

| Chinese dataset | Mean | Variance | Median |
|---|---|---|---|
| Candidate codes per diagnosis. | 13.17 | 8.6 | 10 |
| Reliable evidence per candidate code. | 3 | 2.4 | 2 |
| Tokens per evidence | 43.66 | 52 | 7 |

diagnoses and the ICD codes, We employ the word2vec (Mikolov et al. (2013)) to train the corpus for getting token representations and join them to get the whole representation. Training for Clinical-Simbert took about 156 hours with 1 NVIDIA TITAN RTX GPU with 24 GB memory in total. In the training process, we used learning rate 2e-7, dropout rate 0.1, L2 weight decay 1e3, and batch size of 16 with fp32. The threshold to eliminate unreliable evidence is 0.65 (after normalization). For the inference model based on prompt-tuning, we trained it took about 65 hours with 1 NVIDIA TITNA GPU with 24 GB memory for the Chinese dataset. During prompt-tuning, we used learning rate 1e-4, dropout rate 0.1, L2 weight decay 0.01, and batch size of 16 with fp32.

## 3.4 RESULTS

The results are shown in Table 4. For each baseline, we use the discharge summary in the EMR and the evidence set by the CICD-Coder for all ICD codes in one EMR as input respectively. Firstly, on average, CICD-Coder(discharge summary) is 36.73%, 37.28%, 36.79%, 16.68%, and 8.33% better than other baselines that take the same input in precision, recall, F1, accuracy, and p@5 metrics respectively. This demonstrates the superiority of the prompt-tuning framework in the context of ICD coding tasks. Furthermore, on average, the methods using evidence sets as input improve precision, recall, F1, accuracy, and p@5 by 11.74%, 12.28%, 11.92%, 15.15%, and 25.50%, respectively, compared to the methods using discharge summary as input. These results prove that the discharge summary is not enough to support ICD coding for Chinese EMRs. Simultaneously, the evidence set obtained through prior knowledge retrieval and the evidence evaluation strategy possesses rich and comprehensive clinical information, sufficient to validate the reasonableness of the corresponding candidate ICD codes. Then we compare CICD-Coder(evidence set) with other baselines that take the same input, on average, CICD-Coder(evidence set) is 34.51%, 31.56%, 31.75%, 9.09%, and 5.09% better than other baselines in precision, recall, F1, accuracy, and p@5 metrics respectively. This underscores the superior advantage of our method in the task of automatic ICD coding for Chinese EMRs. Finally, we remove the keywords, sources, and occurrence times added after each piece of evidence in the template design module, and record the versions as "CICD-Coder(without addition)" in Table 4, we can clearly observe the benefits brought by the module.

The significant improvement in macro-averaged metrics indicates that our method performs better in recommending rare or tail ICD codes and the enhanced discriminative ability for distinguishing between confusing ICD codes. This improvement can be attributed to our approach of finding supporting evidence sets under each candidate code axis and reasoning about the overall code support based on this evidence. At the same time, our method also exhibits distinct evidence-based differences for codes that are prone to confusion, ultimately improving code discrimination.

Table 4: Experimental results to verify the effectiveness of CICD-Coder. For each baseline, we use the discharge summary in the EMR and the evidence set by CICD-Coder as input respectively.

| Model | Macro | | | Accuracy | p@5 |
| | Precision | Recall | F1 | | |
|---|---|---|---|---|---|
| LAAT(discharge summary) | 18.06% | 17.81% | 17.94% | 36.65% | 59.65% |
| LAAT(evidence set) | 28.90% | 29.33% | 29.56% | 59.26% | 85.29% |
| jointLAAT(discharge summary) | 22.29% | 22.45% | 22.37% | 41.52% | 62.35% |
| jointLAAT(evidence set) | 33.14% | 35.17% | 34.54% | 63.59% | 89.54% |
| MSMN(discharge summary) | 7.70% | 6.50% | 7.05% | 53.46% | 58.69% |
| MSMN(evidence set) | 29.79% | 32.07% | 30.89% | 68.50% | **94.57%** |
| T5(discharge summary) | 58.53% | 60.89% | 60.05% | 62.64% | 65.27% |
| T5(evidence set) | 63.50% | 64.78% | 64.12% | 69.58% | 81.17% |
| CICD-Coder(discharge summary) | 63.37% | 64.19% | 63.64% | 65.25% | 69.82% |
| CICD-Coder(evidence set) | **73.34%** | **71.90%** | **71.53%** | **74.32%** | 92.73% |
| CICD-Coder(without addition) | 71.12% | 69.07% | 69.78% | 72.45% | 91.90% |

Table 5: Interpretability assessment results of evidence set for the recommended ICD code on 100 Chinese EMRs by ICD coders. The evidence set contains two groups, one is evaluated by the Clinical-Simbert model, and the other is not.

| Degree | Before Clinical-Simbert | After Clinical-Simbert | Variable value |
|---|---|---|---|
| Fully supported | 20.65% | 48.05% | ↑ 27.40% |
| Partially supported | 44.72% | 48.11% | ↑ 3.39% |
| Unable to supported | 34.63% | 3.84% | ↓ 30.79% |

## 3.5 ASSESSMENTS OF INTERPRETABILITY FROM ICD CODERS

To assess the interpretability of the evidence set under all axes for the recommended ICD code, we invite professional ICD coders to evaluate the evidence. We selected 100 EMRs from the test set and asked coders to mark the supporting degree of each evidence set for the corresponding ICD code. We set three levels: fully supported, partially supported, and unable to be supported. Take fully supported for example, which means that when ICD coders see the evidence set, they believe that the corresponding recommended code is correct, and they no longer need to read the entire medical record. The penultimate column in Table 5 is the final result. We can see that nearly half of the evidence set provided can support the corresponding ICD code fully, and only 3.84% of the evidence set is invalid. The ICD coders believe that the provision of evidence makes them more trustful and accepting of CICD-Coder. Even if the evidence is wrong, they can only look for evidence for the corresponding axis, and the overall efficiency is greatly improved.

On the other hand, to evaluate the effectiveness of the Clinical-Simbert model for evidence assessment, we asked ICD coders to mark the supporting degree of each evidence set before assessment, the results are shown in the second column of Table 5. Comparing the second and third columns of Table 5, the fully supported and partially supported levels are increased by 27.4% and 3.38% respectively, and the unable to be supported level is greatly decreased by 30.79%. This proves that our Clinical-Simbert model can effectively evaluate the relevance of evidence and corresponding knowledge under the axes of candidate ICD codes.

## 4 CONCLUSIONS

In this paper, we proposed a Chinese EMR-based ICD coding with multi-axial supported clinical evidence. We developed an evidence retrieval and evaluation module to find the reliable evidence set for the recommended ICD code. We employed the prompt tuning method to determine whether the candidate ICD code can be fully supported by the evidence set. The experimental results demonstrate that our method has achieved state-of-the-art performance and exhibits excellent interpretability.

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
