# A  APPENDIX

## A.1  THE CHINESE VERSION OF FIGURE 3 OF THE PAPER

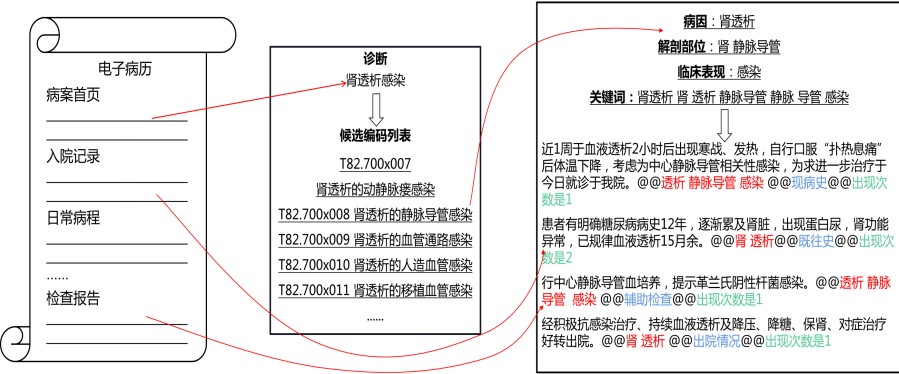

Figure 1: A real example to show the candidate codes of one diagnosis and the reliable evidence under multi-axes for one of the candidate codes (also the correct ICD code), the example is from a real Chinese EMR. Each piece of evidence is followed by a few "@@" symbols for more information helpful for inference, including keywords, the appearing field in the medical record, and times of appearance.

## A.2  DETAILED DESCRIPTION OF EVIDENCE RETRIEVAL

The evidence retrieval module consists of five steps:

1. Extraction of the diagnosis list: Firstly, we extract the diagnosis list from the Chinese electronic medical records. The diagnosis list in Chinese electronic medical records is well-structured and easily extractable.

2. Identification of candidate codes: For each diagnosis, we identify the top N candidate codes. We employ the edit distance, tf-idf, and word2vec-based similarity computation methods to capture the top N candidate codes among all the ICD codes, which are related to each diagnosis at the character, discrimination, and semantic levels, respectively. For the similarity-based method, we use word2vec to train the Chinese EMRs collected and obtain token representations, which are then combined to form the whole representation of the diagnosis and the ICD code. During the experiment, we set the number of candidate codes for each diagnosis to 50. The recall rate of the candidate ICD codes compared to the correct ICD codes for the training set, development set, and test set is 96.58%, 95.43%, and 95.55%, respectively. The recall rate increases with the increase in the number of candidate codes. For example, if the number is increased to 100, the recall rate can exceed 98%. Therefore, during replication, you can adjust the number of candidate codes based on the specific requirements.

3. Determination of retrieval keywords: We utilize a multi-axial parser to categorize the knowledge under each candidate code. Additionally, considering the effectiveness of rich synonym descriptions in ICD coding (as verified by the MSMN method, we incorporate synonyms for these keywords as supplementary information.

4. Greedy evidence retrieval based on keywords and prior knowledge: It is evident that evidence containing more keywords is considered to be more representative and reliable. Based on this, we sort all possible combinations of keywords in descending order according to the number of keywords and retrieve the sentences containing them from likely locations recorded in the prior knowledge. We replace sentences with fewer keywords with more evidence based on a greedy strategy. Once the supporting evidence for a keyword has been retrieved, the retrieval process does not continue for that particular keyword. We stop retrieving when all keywords have been traversed.

5. If you encounter a situation where the evidence retrieval for a specific keyword is unsuccessful in practice, you may consider expanding the retrieval scope to the entire medical record. This is due to

the varying quality of actual EMRs, which may not strictly adhere to standardized writing requirements. The feasibility of this approach depends on the quality of the Chinese electronic medical records available to you. In our actual experiments, such cases were relatively rare, accounting for approximately 15%.

## A.3 ABBREVIATED TABLE OF PRIOR INFORMATION FOR POTENTIAL LOCATIONS OF CLINICAL SUPPORTING EVIDENCE FOR ICD CODES

Table 1: This is a small section of a complete prior knowledge table, listing the possible locations in medical records where clinically supporting evidence for three-digit code groups of interest may be found.

| Code Range and name | Possible locations for evidence appearance in Chinese EMRs | | | |
|---|---|---|---|---|
| A00-A09 Intestinal infectious diseases | Testing (and results) | Symptom presentation | History of exposure to endemic or epidemic areas | |
| A15-A19 Tuberculosis | Symptom presentation | Examination (and results) | Chest examination results | Vaccination and Infectious Disease History |
| A20-A28 Certain zoonotic bacterial diseases | Testing (and results) | Occupation and working conditions | | |
| A30-A49 Other bacterial diseases | Skin and mucous membrane examination results | Testing (and results) | Neurological reflex examination results | |
| A50-A64 Sexually transmitted infections | Skin and mucous membrane examination results | External genital examination results | Testing (and results) | |
| A65-A69 Other spirochetal diseases | Skin and mucous membrane examination results | Testing (and results) | History of exposure to endemic or epidemic areas | |
| A70-A74 Other diseases caused by chlamydia | Chest examination results | Head and facial examination results | Testing (and results) | |
| A75-A79 Rickettsioses | Testing (and results) | History of exposure to endemic or epidemic areas | | |
| ... | ... | ... | ... | ... |