# OpenReview forum: "CICD-Coder: Chinese EMRs Based ICD Coding With Multi-axial  Supported Clinical Evidence"
_ICLR.cc/2024/Conference — Submitted to ICLR 2024_

### Official Review · Reviewer_Lv7G · 2023-10-20

**Soundness:** 3 good
**Presentation:** 3 good
**Contribution:** 2 fair
**Rating:** 6
**Confidence:** 2

**Summary:**

The paper describe a novel method to identify icd code in Chinese clinical notes. The methods include fine retrieving relevant codes and translate the task to binary classification task by feeding a template prompt to a t5 model.

**Strengths:**

The task solved is important
The method described is novel

**Weaknesses:**

Lack of competitive baselines, since the task is novel it’s hard to compare to other methods , hence the validity of the method presented is unclear

**Questions:**

1. Will the data used for evaluation will be publicly avilable?

---

> ### Author Response · Authors · 2023-11-19
> **Reply**
>
> We thank Reviewer Lv7G for reviewing our paper and for the insightful comments. We hope our answers will address the concerns and clarify the contributions of the paper. Additionally, we have revised our submission, and the modified areas are highlighted in blue in the updated submission.
>
>
> **Weaknesses: Lack of competitive baselines, since the task is novel it’s hard to compare to other methods, hence the validity of the method presented is unclear.**
>
> You can find the answer in the “Response for Limitations of Baseline Models in the Experimental Section”.
>
> **Questions:Will the data used for evaluation will be publicly avilable?**
>
> Thank you for your valuable feedback.
> The Chinese electronic medical record (EMR) dataset we utilized in our study was collected from a collaborative research effort involving four different hospitals. As we are conducting joint research with these institutions, the issue of dataset availability needs to be discussed and agreed upon with our collaborative partners. We are actively engaged in these discussions to ensure that any decisions regarding the dataset's public availability align with the interests and privacy considerations of all involved parties.

---

### Official Review · Reviewer_owsX · 2023-10-31

**Soundness:** 2 fair
**Presentation:** 2 fair
**Contribution:** 2 fair
**Rating:** 3
**Confidence:** 4

**Summary:**

This paper introduces the CICD-Coder, a new framework for improving ICD coding in Chinese electronic medical records (EMRs). It addresses challenges in the unique features of Chinese EMRs. The CICD-Coder analyzes crucial disease knowledge, retrieves relevant clinical evidence from the EMRs as the additional features, and finally uses masked language prediction with prompts to perform ICD coding under the support of the retrieved evidence. The experiment results show its effectiveness is significant in a Chinese EMR dataset.

**Strengths:**

1. The paper pioneers the exploration of unique challenges in ICD coding within Chinese electronic medical records (EMRs), an area not extensively covered in existing literature.
2. The authors introduce an innovative evidence retrieval module within their proposed CICD-Coder framework, marking a significant advancement in ICD coding performance. This module stands out for its potential to substantively improve coding accuracy by ensuring that codes are grounded in tangible clinical evidence.

**Weaknesses:**

1. The paper overlooks crucial existing research, particularly the study outlined in [1]. That seminal work similarly employs prompt-based mask prediction for ICD code probability, and its omission here represents a significant gap in the literature review.
2. The absence of comprehensive ablation studies is a notable weakness. The paper would greatly benefit from detailed analyses that demonstrate the specific contributions and impact of the proposed mask-based prediction methodology.
3. The explanation of key methodologies, especially the evidence retrieval module, is vague. Given its critical role in enhancing ICD coding performance, a more in-depth discussion of its design and functionality is essential for readers to fully understand and replicate the study.
4. Certain claims appear unsubstantiated, creating potential confusion. For instance, the assertion regarding the brevity of diagnoses in Chinese EMRs contradicts common characteristics seen in datasets like MIMIC. This discrepancy necessitates clarification to maintain the paper's credibility.
5. The paper exhibits limited novelty when viewed against the backdrop of existing studies like [1]. The underdeveloped evidence retrieval module further diminishes the perceived innovativeness of the CICD-Coder framework. A more thorough exploration of these elements could help underline the unique contributions of the current study.

[1] Yang Z, Wang S, Rawat B P S, et al. Knowledge Injected Prompt Based Fine-tuning for Multi-label Few-shot ICD Coding[C]//Proceedings of the Conference on Empirical Methods in Natural Language Processing. Conference on Empirical Methods in Natural Language Processing. NIH Public Access, 2022, 2022: 1767.

**Questions:**

Will the dataset be available? If not, have you considered applying the proposed method to some public datasets e.g. MIMIC-III and IV? If not, what makes it impossible?

---

> ### Author Response · Authors · 2023-11-19
> **Reply**
>
> We thank Reviewer owsX for reviewing our paper and for the insightful comments. We hope our answers will address the concerns and clarify the contributions of the paper.
>
> **Weaknesses1: The paper overlooks crucial existing research...**
>
> Thank you for your valuable feedback.
>
> Prompt tuning is indeed an effective framework for integrating downstream tasks with pretraining, and it has been widely adopted and utilized in the field of automated ICD coding in recent years.  In the original manuscript, we discussed this point and cited a paper that uses this framework, namely "Multi-Label Few-Shot ICD Coding as Autoregressive Generation with Prompt" published in AAAI 2023. We have taken your feedback into account and have now included the study you mentioned in our revised manuscript.
>
> **Weaknesses3: The explanation of key methodologies, especially the evidence retrieval module, is vague...**
>
> In response to your concern regarding the explanation of key methodologies, particularly the evidence retrieval module, we have provided a more detailed discussion in the appendix to enhance the understanding and replicability of our study.
>
> **Weaknesses4：Certain claims appear unsubstantiated, creating potential confusion...**
>
> To address this concern and substantiate our claim, we conducted a thorough comparison between Chinese and English EMR datasets. Specifically, we collected data from Chinese EMRs and analyzed the average length of diagnostic descriptions and discharge summary. Our findings revealed that the average length of Chinese diagnostic descriptions was 3.74 words, whereas the average length of the discharge summary was 731.80 words. In contrast, when using the widely used MIMIC-III dataset for English EMRs, we observed that the average length of English diagnostic descriptions was 14.73 words, and the average length of discharge summary was 1600.36 words. To provide further evidence, we have included more comprehensive statistical information in Table 1 of the paper.
>
> Based on these extensive statistics, it is evident that diagnostic descriptions in Chinese EMRs are significantly briefer and lack sufficient information compared to their English counterparts. Moreover, discharge summary in Chinese EMRs are also shorter and insufficient to support automated ICD coding. We believe that these additional details and comparisons will enhance the credibility and clarity of our paper.
>
> **Weaknesses5：The paper exhibits limited novelty when viewed against the backdrop of existing studies like [1]...**
>
>
> Our research primarily focuses on designing an automatic ICD coding framework that addresses the challenges posed by the characteristics of Chinese electronic medical records (EMRs). In Chinese EMRs, diagnoses are often concise, and discharge summary may lack sufficient information, resulting in scattered evidence supporting ICD coding throughout the entire medical record. It is in response to these characteristics that we have designed a methodological system that combines a domain-specific knowledge-driven evidence retrieval module with a multi-axes evidence-supported ICD coding inference module.
>
> Certainly, there is room for further optimization of each specific technical module in our approach. The retrieval module we have developed is primarily based on the characteristics of Chinese EMRs, as well as the prior knowledge and expert experience related to diseases. Through the evidence evaluation module, the retrieved evidence has already demonstrated its effectiveness in supporting ICD coding inference and providing good interpretability. Therefore, we did not pursue further detailed optimization in this particular aspect. However, we appreciate your suggestion, as further optimization could potentially lead to performance improvements. The main focus of this study is to propose a framework that is more suitable for automatic ICD coding in Chinese EMRs. Subsequent work can delve into exploring each module in more detail.
>
>
> **Question1：Will the dataset be available? ...**
>
>
> The Chinese electronic medical record (EMR) dataset we utilized in our study was collected from a collaborative research effort involving four different hospitals. As we are conducting joint research with these institutions, the issue of dataset availability needs to be discussed and agreed upon with our collaborative partners. We are actively engaged in these discussions to ensure that any decisions regarding the dataset's public availability align with the interests and privacy considerations of all involved parties.
>
> While we recognize the value of applying our method to English EMRs, it would require significant modifications and adaptations to account for the differences in data structure and language characteristics. Exploring the application of our method to English EMRs is an intriguing avenue for future research, and we appreciate your suggestion in this regard.

---

### Official Review · Reviewer_uUev · 2023-11-03

**Soundness:** 3 good
**Presentation:** 3 good
**Contribution:** 2 fair
**Rating:** 5
**Confidence:** 3

**Summary:**

This paper proposes a new automatic ICD coding framework, namely CICD-Coder, for the Chinese Electronic Medical Records (EMRs). The presented framework utilizes multi-axes knowledge of the disease and retrieves clinical evidence from the EMRs. The work primarily focused on extracting ICD codes from Chinese EMR, which poses additional constraints and problems that need to be tackled in addition to extracting codes from English EMR. Experiments are conducted on real, Chinese dataset and evaluated by ICD coders as well.

**Strengths:**

* The paper is well-motivated in the sense that it attempts to address challenges arising from Chinese EMR that is not often present in English EMR. While I am not very familiar with existing works on Chinese EMR, this line of work might be interesting for practitioners and ML community that is interested in Chinese EMRs.
* Experiments are conducted thoroughly, using both Chinese datasets and assessments made by ICD coders.
* The proposed method does show improved performances compared with the baselines. However, there are some other concerns (see below).

**Weaknesses:**

1. While the proposed method in the paper does improve over its baselines. The improvement does not have a large margin and is not significant.
2.  Also, the number of baselines is too small. There are only two baselines which makes the comparison and results non-exhaustive. Unless there is a strong reason and explanation for using only two baselines, the authors could consider using more.
3. In 2.2.1, the author says that **We have prior information about the likely appearance location of the supporting evidence in the medical record by professional ICD coders**. In the last sentence, they say that this prior information makes the process **fast and efficient**. Given that the paper is application-oriented, I wonder what would happen when practitioners do not have this prior information. It seems to be that having such prior information is a strong assumption.
4. Minor issue: part of the paper contains grammar issues.

In particular, 1 and 2 are the primary reasons for lowering the score below the acceptance threshold. If they are properly addressed, I might consider to raise my score.

**Questions:**

Given that the paper's focus is on Chinese EMRs, I am not sure whether some setups made in the paper is relevant or specific to Chinese EMR. For instance, for the two principles made in the paper, the first principle is **The source of the evidence limits what the evidence can describe and how credible it is.**, and the second principle is **The number of repeating times of one piece of evidence in the Chinese EMR determines its importance**. I am not sure whether those two principles are specific to Chinese EMRs. It seems to me that those two principles could broadly apply to EMRs in most countries. Thus, it is debatable whether the design of the principles is specific to Chinese EMR, which makes those two design principles seem a bit too general. I wonder whether the authors have any idea about the specific design of the principles.

---

> ### Author Response · Authors · 2023-11-19
> **Reply**
>
> We thank Reviewer uUev for reviewing our paper and for the insightful comments. We hope our answers will address the concerns and clarify the contributions of the paper.
>
> **Weaknesses1: While the proposed method in the paper...**
>
> Firstly, we would like to emphasize that ICD coding is a challenging task due to the large number of codes involved. Therefore, achieving significant improvements in performance metrics can be difficult.
>
> In our study, we compared our proposed method, CICD-Coder, with baselines that also take the evidence set as input. On average, our method outperforms these baselines by 26.70\%, 23.48\%, 24.03\%, 5.28\%, and 4.86\% in precision, recall, F1, accuracy, and p@5 metrics, respectively. The significant improvement in macro-averaged metrics indicates that our method performs better in recommending rare or tail ICD codes and the enhanced discriminative ability for distinguishing between confusing ICD codes. This improvement can be attributed to our approach of finding supporting evidence sets under each candidate code axis and reasoning about the overall code support based on this evidence. At the same time, our method also exhibits distinct evidence-based differences for codes that are prone to confusion, ultimately improving code discrimination.
>
> Regarding the lower performance of our method compared to MSMN in the p@5 metric, we would like to clarify that MSMN performs classification within the entire ICD code label space, while our method performs binary classification for each candidate code individually within the label space of all recalled candidates. Therefore, the upper limit of our method's performance in p@5 is essentially the recall rate of the candidate codes. The recall rate for candidate codes is approximately 95\%-96\%. Consequently, this limitation can explain the difference in performance for p@5 compared to MSMN. However, there is a solution to address this issue. The recall rate of candidate codes can be effectively improved by increasing the number of candidate codes.
>
> **Weaknesses3: In 2.2.1, the author says that We have prior information ......**
>
>
> Thank you for your valuable feedback regarding the assumption of prior information in our paper. To address your concern, we provided a detailed explanation in the first paragraph of section 2.2.2 regarding the acquisition method of prior knowledge.
>
> In the retrieval module, this prior information plays a crucial role in improving the efficiency of evidence retrieval. With prior knowledge, we can narrow down the search scope from the entire EMR to the likely appearance locations, and only resort to searching the entire medical record when the retrieved evidence is insufficient. We have estimated that the evidence retrieval strategy empowered by this prior knowledge can achieve a speed improvement of approximately 18 times compared to directly searching the entire medical record.
>
> In terms of reliability, evidence retrieval based on the entire medical record may yield a higher recall rate of reliable evidence. However, it is indeed a trade-off between recall and reliability. We have evaluated that, under the same number of supporting pieces of evidence, the evidence retrieval strategy based on prior knowledge can achieve approximately 33\% higher reliability of the evidence set compared to the strategy without prior knowledge.
>
>
> **Weaknesses4: Minor issue: part of the paper contains grammar issues.**
>
>
> In the revised version that we have submitted, we have conducted a thorough grammar check and made the necessary corrections throughout the paper.
>
> **Questions1: Given that the paper's focus is on Chinese EMRs, ...**
>
>
> Thank you for your thoughtful comments and suggestions. We agree that the two principles mentioned in our paper could potentially apply to electronic medical records in other languages as well, and yield benefits.
>
> The reason we emphasized these principles in the context of Chinese EMRs is that most of the existing automatic ICD coding methods were originally developed for English EMRs. We found that the length of discharge summary in English EMRs is more than twice that of Chinese EMRs. As a result, many automatic ICD coding methods for English EMRs directly use the discharge summary as input. However, in Chinese EMRs, the supporting evidence for each ICD code may be scattered throughout different sections of the medical record. Therefore, the source and frequency of appearance of each piece of evidence become meaningful.
>
> By incorporating these two pieces of information into our prompt, we aim to provide additional valuable information for determining whether a set of evidence can support the reasoning task of candidate code selection. Based on the ablative experiments in the last two rows of Table 3, it is indeed observable that incorporating this information can lead to performance improvements.

---

### Official Review · Reviewer_1CT8 · 2023-11-06

**Soundness:** 1 poor
**Presentation:** 1 poor
**Contribution:** 1 poor
**Rating:** 1
**Confidence:** 4

**Summary:**

The authors of this study concentrate on the utilization of Chinese Electronic Medical Records (EMRs) for ICD coding. They emphasize the complexity of Chinese EMRs, where information is scattered across various locations, necessitating a comprehensive approach for ICD prediction. To enhance the precision of their predictions, the authors propose incorporating additional evidence after the initial ICD prediction. They accomplish this by training a retrieval model designed to retrieve the relevant evidence corresponding to a given diagnosis. Their findings suggest an improvement in ICD coding accuracy on Chinese datasets, albeit with limited exploration. Nevertheless, the paper is deemed in need of substantial revision, and in my opinion, it is not yet ready for publication.

**Strengths:**

Retrieval seems to be important for ICD coding - The importance of retrieval in ICD coding is underscored by this paper, which emphasizes the potential for enhanced ICD coding accuracy through the retrieval of relevant evidence. However, it is worth noting that the execution of these ideas within the paper falls short in terms of clarity and effectiveness.

**Weaknesses:**

**Poor presentation of ideas**

- The paper exhibits a deficiency in the presentation of its ideas, lacking a comprehensive methodological description that is essential for clarity and reproducibility. This shortcoming necessitates substantial reworking before the manuscript can be considered for publication.

**Lack of Motivation**

- The motivation for focusing on Chinese EMRs in the study is inadequately substantiated. The problems highlighted by the authors are not unique to Chinese EMRs and are also prevalent in other languages, including English. A stronger rationale is required to establish the relevance and significance of this specific focus.

**Poor experimentation**

- The paper's experimental approach is notably deficient as it omits comparisons with other relevant baselines. For example, it does not include comparisons with established methods like LAAT and other techniques known for their efficacy in ICD coding. This omission hinders the paper's ability to demonstrate its effectiveness and distinguish itself within the field. Addressing this issue is imperative to improve the paper's readiness for publication.

**Questions:**

1. ***In Section 2.2.1*** the authors mention that they obtain the ICD codes. Which method is used to obtain the initial ICD codes.
2. ***Section 2.2.1*** mentions the use of prior knowledge to identify likely evidence for different icd codes. Is there a comprehensive list of these rules? The authors do not mention the rules

---

> ### Author Response · Authors · 2023-11-19
> **Reply**
>
> We thank Reviewer 1CT8 for reviewing our paper and for the insightful comments. We hope our answers will address the concerns and clarify the contributions of the paper.
>
> **Question1: In Section 2.2.1 the authors mention that they obtain the ICD codes. Which method is used to obtain the initial ICD codes.**
>
> Thank you for your comment. We appreciate your observation that there is a paragraph below Section 2.2 and above Section 2.2.1 that explains how we obtain the candidate codes, but it lacks a specific subheading. To improve the overall logical flow and organization of the paper, we have now added a new subheading, "2.2.1: Obtaining Candidate Codes for Each Diagnosis," to address the method used to obtain the initial ICD codes.
>
>
> **Question2：Section 2.2.1 mentions the use of prior knowledge to identify likely evidence for different icd codes. Is there a comprehensive list of these rules? The authors do not mention the rules.**
>
> Thank you for your valuable feedback.
>
> The prior knowledge we utilized was obtained through discussions with experienced professional coders who work directly with ICD codes. Specifically, the ICD coding system categorizes all codes into 212 categories at the three-digit level[1]. Through our discussions, we determined that each category focuses on similar likely appearance locations of the supporting evidence in the medical record. We summarized a total of 88 possible locations where evidence may appear in Chinese EMRs. Furthermore, we asked the ICD coders to annotate the possible locations of each code category, specifically focusing on the evidence that may appear. This process resulted in the formation of a prior information table.
>
> To provide you with a clearer understanding of this prior knowledge, we have included a small excerpt from the full table in the appendix.
>
> [1] Buck C J. 2017 ICD-10-CM Hospital Professional Edition-E-Book[M]. Elsevier Health Sciences, 2016.
>
> **Weaknesses1: Poor presentation of ideas**
>
> Thank you for your feedback on our paper.
> Our research primarily focuses on designing an automatic ICD coding framework that addresses the challenges posed by the characteristics of Chinese electronic medical records (EMRs). In Chinese EMRs, diagnoses are often concise, and discharge summary may lack sufficient information, resulting in scattered evidence supporting ICD coding throughout the entire medical record. In response to these characteristics, we have designed a methodological system that combines a domain-specific knowledge-driven evidence retrieval module with a multi-axes evidence-supported ICD coding inference module.
>
>
>
> **Weaknesses2: Lack of Motivation**
>
> Thank you for providing your valuable feedback. While we acknowledge that the problems we highlighted are not unique to Chinese EMRs and can be found in other languages, including English, we believe that these issues are more pronounced in Chinese EMRs due to the specific characteristics of the Chinese language.
>
> To support our claim, we conducted a comparison between Chinese and English EMR datasets. We collected data from Chinese EMRs and found that the average length of Chinese diagnostic descriptions was 3.74 words, while the average length of the discharge summary was 731.80 words. In contrast, using the commonly used MIMIC-III dataset for English EMRs, we found that the average length of English diagnostic descriptions was 14.73 words, and the average length of discharge summary was 1600.36 words. We have provided more detailed statistical information in Table 1 of the revised manuscript.
>
> Based on the aforementioned statistics, it is evident that diagnostic descriptions in Chinese EMRs are significantly shorter and lack sufficient information. Moreover, discharge summary are also shorter and insufficient to support automated ICD coding. Additionally, clinical evidence supporting ICD coding across different axes is scattered throughout the entire Chinese EMR, making existing automated ICD coding methods that are effective in English EMRs less effective in Chinese EMRs.
>
> Therefore, we believe there is a strong motivation to propose a specialized automatic ICD coding method for Chinese EMRs.

---

### Author Response · Authors · 2023-11-19
**Response for Limitations of Baseline Models in the Experimental Section**

Dear Reviewers,

Thank you for your valuable feedback regarding the experimental approach in our paper. We appreciate your suggestions and acknowledge the importance of comparing our proposed method with other relevant baselines to demonstrate its effectiveness and distinguish it within the field.

In our original manuscript, we compared our approach with a strong baseline, namely the MSMN method, which was a state-of-the-art technique proposed in 2022. However, we understand your concern and agree that it is essential to provide a more comprehensive comparison. Therefore, we have added additional baseline comparison experiments. The table below presents the results of the most recently completed experiment.

Table: Experimental results to verify the effectiveness of CICD-Coder. For each baseline, we use the discharge summary in the EMR and the evidence set by CICD-Coder as input respectively. We remove the keywords, sources, and occurrence times added after each piece of evidence in the template and record this version as "CICD-Coder (without addition)".

| Model | p@5 | Precision | Recall | Accuracy | F1 |
| --- | --- | --- | --- | --- | --- |
| LAAT(discharge summary) | 18.06% | 17.81% | 17.94% | 36.65% | 59.65% |
| LAAT(evidence set) | 28.90% | 29.33% | 29.56% | 59.26% | 85.29% |
| jointLAAT(discharge summary) | 22.29% | 22.45% | 22.37% | 41.52% | 62.35% |
| jointLAAT(evidence set) | 33.14% | 35.17% | 34.54% | 63.59% | 89.54% |
| MSMN(discharge summary) | 7.70% | 6.50% | 7.05% | 53.46% | 58.69% |
| MSMN(evidence set) | 29.79% | 32.07% | 30.89% | 68.50% | 94.57% |
| T5(discharge summary) | 58.53% | 60.89% | 60.05% | 62.64% | 65.27% |
| T5(evidence set) | 63.50% | 64.78% | 64.12% | 69.58% | 81.17% |
| CICD-Coder(discharge summary) | 63.37% | 64.19% | 63.64% | 65.25% | 69.82% |
| CICD-Coder(evidence set) | 73.34% | 71.90% | 71.53% | 74.32% | 92.73% |
| CICD-Coder(without addition) | 71.12% | 69.07% | 69.78% | 72.45% | 91.90% |

As shown in the above table, after incorporating LAAT, jointLAAT, and CICD-Coder(discharge summary), the following experimental conclusions can be obtained.

- On average, CICD-Coder(discharge summary) is 36.73\%, 37.28\%, 36.79\%, 16.68\%, and 8.33\% better than other baselines that take the same input in precision, recall, F1, accuracy, and p@5 metrics respectively. This demonstrates the superiority of the prompt-tuning framework in the context of ICD coding tasks.
- On average, the methods using evidence sets as input improve precision, recall, F1, accuracy, and p@5 by 11.74\%, 12.28\%, 11.92\%, 15.15\%, and 25.50\%, respectively, compared to the methods using discharge summary as input. These results prove that the discharge summary is not enough to support ICD coding for Chinese EMRs. Simultaneously, the evidence set obtained through prior knowledge retrieval and the evidence evaluation strategy possesses rich and comprehensive clinical information, sufficient to validate the reasonableness of the corresponding candidate ICD codes.
- We compare CICD-Coder(evidence set) with other baselines that take the same input, on average, CICD-Coder(evidence set) is  34.51\%, 31.56\%, 31.75\%, 9.09\%, and 5.09\% better than other baselines in precision, recall, F1, accuracy, and p@5 metrics respectively. This underscores the superior advantage of our method in the task of automatic ICD coding for Chinese EMRs.

We hope that our latest experimental results can provide a more comprehensive and thorough demonstration of the effectiveness of our proposed method for automatic ICD coding in Chinese EMRs.  We hope that these revisions and additional insights will address your concerns and help you to fully appreciate the significance and potential impact of our research. Thank you for considering our work.

Best regards,

Authors

---

### Author Response · Authors · 2023-11-22

Dear Reviewers,

We appreciate your valuable comments and suggestions. In response to the concerns you've raised, we have made substantial revisions to our manuscript and its appendices, aiming to provide a clearer explanation and address your queries. Here are the details of the changes we've made:

1. **Enhanced clarity and reproducibility of the retrieval module:** In line with your feedback, we've strived to provide a more understandable and reproducible description of the evidence retrieval module, which leverages prior knowledge and multi-axial knowledge of ICD codes. The following changes have been made:

   - A new subheading (2.2.1) has been added to the section that describes the method for finding candidate codes.
   - More detailed retrieval steps have been provided in Appendix A.2.
   - A partial table of examples of prior knowledge used for retrieval in Appendix A.3.
   - In the revised section 2.2.2, we have included detailed methods for acquiring prior knowledge.

2. **Statistical support for the characteristics of Chinese EMRs:** To substantiate our claim that "In Chinese EMRs, diagnoses are often concise, and the discharge summary may lack sufficient information, resulting in scattered evidence supporting ICD coding throughout the entire medical record," we have presented the relevant statistical results in Table 1 of the revised manuscript. This statistical work was completed at the very outset of our project and has formed the basis for our subsequent work. Due to space constraints, these results were not included in the original manuscript.

   - Our data from Chinese EMRs shows that the average length of Chinese diagnostic descriptions is 3.74 words, while the average length of the discharge summary is 731.80 words.
   - In comparison, using the commonly used MIMIC-III dataset for English EMRs, we found that the average length of English diagnostic descriptions is 14.73 words, and the average length of discharge summary is 1600.36 words.


We are confident that these revisions have not only addressed your concerns but also improved the clarity and quality of our paper. Furthermore, we believe that these enhancements will aid in your full appreciation of the significance and potential impact of our research.

Best regards,

Authors

---

### Meta-Review · Area_Chair_Vr91 · 2023-12-02

**Metareview:**

I agree with the comments of the reviewers. I believe their insights will be valuable in assisting the authors to improve their paper.

**Justification For Why Not Higher Score:**

NA

**Justification For Why Not Lower Score:**

NA

---

### Decision · Program_Chairs · 2024-01-16

Reject